# Breath Tests Used in the Context of Bariatric Surgery

**DOI:** 10.3390/diagnostics12123170

**Published:** 2022-12-15

**Authors:** Daniel Karas, Marek Bužga, David Stejskal, Petr Kocna, Pavol Holéczy, Adéla Novotná, Zdeněk Švagera

**Affiliations:** 1Institute of Laboratory Medicine, Faculty of Medicine, University of Ostrava, Syllabova 19, 703 00 Ostrava, Czech Republic; 2Department of Human Movement Studies, Faculty of Education, University of Ostrava, Fráni Šrámka 3, 709 00 Ostrava, Czech Republic; 3Department of Physiology and Pathophysiology, Faculty of Medicine, University of Ostrava, Syllabova 19, 703 00 Ostrava, Czech Republic; 4Institute of Laboratory Medicine, University Hospital Ostrava, 17. Listopadu 1790/5, 708 52 Ostrava, Czech Republic; 5Institute of Medical Biochemistry and Laboratory Diagnostics, 1st Faculty of Medicine, Charles University in Prague, Kateřinská 1660/32, 121 08 Prague, Czech Republic; 6Department of Surgery, Vítkovice Hospital, Zalužanského 1192/15, 703 00 Ostrava, Czech Republic; 7Department of Surgical Disciplines, Faculty of Medicine, University of Ostrava, Syllabova 19, 703 00 Ostrava, Czech Republic; 8Department of Epidemiology and Public Health, Faculty of Medicine, University of Ostrava, Syllabova 19, 703 00 Ostrava, Czech Republic

**Keywords:** organs, substances administered, gases determined, breath tests, bariatric surgery, obesity

## Abstract

This review article focuses on the use of breath tests in the field of bariatrics and obesitology. The first part of the review is an introduction to breath test problematics with a focus on their use in bariatrics. The second part provides a brief history of breath testing. Part three describes how breath tests are used for monitoring certain processes in various organs and various substances in exhaled air and how the results are analyzed and evaluated. The last part covers studies that described the use of breath tests for monitoring patients that underwent bariatric treatments. Although the number of relevant studies is small, this review could promote the future use of breath testing in the context of bariatric treatments.

## 1. Introduction

As modern diagnostic tools, breath tests are rapid, simple, and non-invasive [1,2]. On the other hand, it may be problematic to acquire samples of exhaled air from patients over relatively long periods. Breath tests are used to evaluate the functions of several organs (e.g., liver, pancreas, stomach, intestine), or rather, the processes and phenomena that take place in those organs. The evaluations require measurements of certain gases in exhaled air after administering a specific substrate to the patient.

The usage of breath tests must meet European guidelines on the indications, performance, and clinical impact of hydrogen and methane breath tests in adult and pediatric patients: European Association for Gastroenterology, Endoscopy and Nutrition, European Society of Neurogastroenterology and Motility, and European Society for Paediatric Gastroenterology Hepatology and Nutrition consensus from 2022 [3].

Table 1 summarizes the features of breath tests in current use, i.e., the processes/phenomena that can be detected in the organism, the organs that can be tested with breath tests, and the substances used for detection. Table 1 also mentions how substrates are administered, what methods are used for detection, and how the results are evaluated.

Breath tests have potentially wide applications in the fields of bariatrics and obesitology. Nevertheless, until recently, they have been scarcely used in these fields. This fact was evident in the small number of relevant studies we identified that were published in the last three decades (1990–2021). Currently, various invasive methods are typically used to monitor patient conditions after bariatric surgery. Therefore, applications for breath tests in bariatrics could be of particular importance because they are non-invasive.

## 2. Breath Tests

### 2.1. History of Breath Tests

The breath test is a non-invasive method that focuses on the analysis of exhaled air. Its importance dates back to antiquity when it was known that certain characteristic mouth odors could be associated with specific diseases [6].

The modern breath test was introduced in the early 1960s. The first studies used gas chromatography to measure increases in the hydrogen concentration in exhaled air after the consumption of baked beans [7] and expired ^14^C after the consumption of ^14^C-labeled fats [8]. In the 1970s, breath tests were introduced in laboratory diagnostics. Those tests measured hydrogen (H_2_) and ^13^C- or ^14^C-labeled carbon dioxide (CO_2_) in exhaled air [9]. Specifically, the stable carbon isotope, ^13^C, was established as a marker by Lacroix in 1973 [10]. Two decades later, a breath test was discovered, which could be used to diagnose *Helicobacter pylori* based on the administration of ^13^C/^14^C-labeled urea. That test initiated a significant expansion in the use of ^13^C and ^14^C tests in clinical practice [11,12]. However, due to its radioactive nature, ^14^C carbon is rarely used in current practices. In the 1990s, nitric oxide (NO) was discovered as a marker that could be detected in a breath test. The exhaled fraction of NO in exhaled air, called fractional exhaled nitric oxide (FeNO), is currently used in clinical practice to diagnose and treat asthma [13,14]. NO is also a biomarker in differential diagnosis to distinguish between a peptic ulcer in the stomach and non-ulcer-related dyspepsia [15].

### 2.2. Description of Breath Tests

Currently, breath tests are most commonly used to measure hydrogen, produced in the colon, together with carbon dioxide, during the anaerobic hydrolysis of carbohydrates. Bacteria metabolize hydrogen to methane and hydrogen sulfide (H_2_S) [16]. The gases produced in the intestine are transported into the blood and exhaled in the air [17,18]. In hydrogen tests, in addition to methane and CO_2_, O_2_ is measured to correct for the alveolar gas concentration. Different breath tests are used to measure specific ^13^CO_2_ and ^14^CO_2_ markers and FeNO [13]. A novelty in the field of breath tests is the ability to determine expired H_2_S [19]. Currently, more than 1000 volatile organic compounds (VOCs) have been evidenced in exhaled air [20].

There are data on the normal range of each substance in exhaled air. When that range is exceeded, it indicates a problem with a particular organ or process. For hydrogen breath tests, a positive result is typically considered a 10–20 ppm increase in concentration over the baseline value measured before the substrate is administered [21]. To evaluate the amount of ^13^C in exhaled air, the δ^13^C value is determined, which indicates the ^13^C/^12^C ratio. This is compared to the internationally accepted Vienna Pee Dee Belemnite standard, which provides the precise amount of ^13^C (1.11237%) expected under normal conditions [11]. The amount of ^13^C glucose in exhaled air is expressed as the so-called C_120_ value, which corresponds to the ^13^C excretion rate (mmol/h) relative to the body surface area at 120 min after the substrate is administered [22].

The substances in exhaled air must be analyzed with the appropriate methods. To analyze hydrogen in exhaled air, there are simple portable analyzers, hand-held battery detectors with electrochemical sensors, analyzers for home use, and analyzers for laboratory use that determine not only hydrogen but also methane [23,24].

The best analytical method for determining the ^13^CO_2_/^12^CO_2_ ratio in exhaled air is called isotope ratio mass spectrometry. Other options include nondispersive infrared spectroscopy [25,26], molecular correlation spectroscopy, which provides continuous measurements with a nasal probe, and laser-assisted ratio analysis [27]. Recent technologies currently undergoing testing include laser absorption, hollow waveguides, capillary absorption spectroscopy, tunable diode laser absorption spectroscopy, and cavity ring-down spectroscopy [28,29].

VOCs can be separated and identified with various methods, including gas chromatography for separation, proton transfer reaction-mass spectrometry; selected ion flow tube mass spectrometry (SIFT-MS), and modern nanoarray technology for identification [30,31]. For example, VOC analysis with gas chromatography, in connection with SIFT-MS, can be performed to determine a differential diagnosis for irritable bowel syndrome and idiopathic bowel disease [32,33].

## 3. Search Strategy and Selection Criteria

To investigate the use of breath tests in the field of bariatric treatments for obesity, we conducted a search of scientific articles published from 1990 to 2021 with the EBSCO Discovery Service tool. We searched all electronic information resources available to our university. Specifically, we searched MEDLINE, Academic Search Complete, Academic Search Ultimate, and CINAHL Plus databases. We searched for the following keywords: “breath tests”, “methacetin breath test”, “bariatrics”, “obesitology”, “obesity”, “metabolic syndrome”, and “basal metabolism”.

In the first phase, we searched for the keywords in the titles of articles; then, we extended the search to the abstracts of the articles. Based on these results, the most relevant publications were selected, and a total of six were found. The search procedure is summarized in Figure 1.

### Application of Breath Tests in Bariatrics

Several studies have described the use of breath tests in different types of bariatric procedures. These procedures include the Roux-en-Y gastric bypass (RYGB), one anastomosis gastric bypass (OAGB), sleeve gastrectomy (SG), biliopancreatic diversion with duodenal switch (BPD/DS), and jejunoileal bypass procedure (JIBP). The studies are summarized in Table 2.

Among those studies, Coelho et al. (2019) and Mouillot et al. (2020) observed bacterial overgrowth in the small intestine (SIBO) of patients with bariatric disorders [34,39]. In a retrospective study by Coelho et al., patients that underwent the RYGB alone later exhibited changes in gastrointestinal tract anatomy, which could lead specifically to the colonization of fermentative bacteria in the duodenum and jejunum. In that study, 18 patients were followed up with the hydrogen breath test, and the test was positive in seven cases [34]. Moreover, Coelho et al. (2019) mentioned four other studies that described breath tests in patients that required SIBO monitoring [34].

Ishida et al. (2007) performed a retrospective study on the hydrogen lactulose breath test for monitoring bacterial overgrowth in 37 patients that had undergone an RYGB. They found that 40.5% of the patients had positive tests [35]. Interestingly, a retrospective study by Lakhani et al. (2008) found that hydrogen breath tests performed after oral glucose showed positive bacterial overgrowth in all patients after an RYGB [36]. Bacterial overgrowth is a known cause of malabsorption. In this type of breath test, bacterial overgrowth in the upper intestine is detected by an increase in exhaled hydrogen (or methane), above 10 parts per million (ppm), at ≤30 min after oral glucose administration. However, when hydrogen increases to at least 10 ppm at ≥45 min after glucose ingestion, it is a sign of malabsorption. In a retrospective study by Andalib et al. (2015), breath tests for detecting SIBO were performed after patients underwent an RYGB [37]. They found that 51 (81%) of 63 patients showed increases in hydrogen (or methane) above 10 ppm at ≥45 min after RYGB, which indicated malabsorption. Of these 51 patients, 46 (90%) also showed an increase in hydrogen (or methane) concentration above 10 ppm at ≤30 min after glucose ingestion, which indicated SIBO [37].

A prospective study by Sabate et al. (2015) monitored SIBO in obese patients before and after bariatric surgery (RYGB or adjustable gastric banding, AGB) with a hydrogen breath test performed after glucose administration. Before bariatric surgery, the breath test was positive in only 15.4% of patients. After surgery, the breath test was positive in 10% of patients that received AGB and 40% of patients that received an RYGB [38].

A retrospective cohort study by Mouillot et al. (2020) included data from 101 patients: 63 had undergone an RYGB, 31 had undergone an OAGB, and 7 had undergone an SG. The glucose breath test was performed to monitor SIBO. They found positive results in 83% (84) of patients, and no significant differences were observed between the different types of bariatric procedures [39].

In a retrospective study by Mathur et al. (2016), gas chromatography of exhaled air was used to monitor the amount of methane and hydrogen produced by gut bacteria in patients after an RYGB or SG. Out of 156 patients, only 13 showed positive results for both hydrogen and methane [40].

Westerink et al. (2020) also studied patients after an RYGB in an observational study. They monitored lactose malabsorption with a hydrogen breath test after lactose administration. They compared 84 patients that were preparing for the RYGB procedure to 84 patients that had undergone the procedure. Lactose malabsorption was observed in 17.9% (N = 15) of patients before bariatric surgery and in 29.8% (N = 25) of patients after surgery [41].

In their prospective, observational, cross-sectional, comparative study, Uribarri-Gonzalez et al. (2021) assessed and compared pancreatic function, digestive dynamics, and nutrient absorption after bariatric surgery with the ^13^C-mixed triglyceride breath test in 95 patients treated with SG (N = 23), RYGB (N = 36), or biliopancreatic diversion with duodenal switch (BPD/DS, N = 36) [42]. The breath test was evaluated by calculating the 6 h ^13^C-cumulative recovery rate (^13^C-CRR), the ^13^C exhalation peak, and the 1 h maximal ^13^C-CRR [42]. They found that the 6 h ^13^C-CRR was significantly reduced after BPD/DS but not after SG or RYGB. Exocrine pancreatic insufficiency (EPI) was defined as a 6 h ^13^C-CRR below the fifth percentile of the control group. EPI was observed in 75% of patients after BDP/DS, 8.3% of patients after RYGB, and 4.3% of patients after SG. Compared to the control group of obese individuals without bariatric treatment, digestion and nutrient absorption occurred earlier in the SG group and later in the RYGB and BPD/DS groups [42].

Venturi et al. (1994) considered a different bariatric procedure from those mentioned in the previous studies, namely the JIBP. They performed a hydrogen breath test after lactulose administration to investigate why abdominal bloating symptoms persisted in a large number of patients after JIBP treatments for morbid obesity. Abdominal bloating can trigger an overgrowth of hydrogen-producing colonizing bacteria in the small intestine. They found that hydrogen production was >100 ppm in 9 of 30 patients [43].

## 4. Discussion

In the context of bariatric treatments, bacterial overgrowth was determined with a hydrogen breath test after lactulose administration in studies by Coelho et al. (2019) [34] and Venturi et al. (1994) [43] and with a hydrogen breath test after glucose administration in a study by Mouillot et al. (2020) [39]. In the study by Coelho et al. (2019), the bariatric intervention was RYGB, and the breath test revealed that 38.8% of patients had SIBO [34]. In the study by Venturi et al. (1994), the intervention was JIBP, and 30% of patients had SIBO [43]. However, in the study by Mouillot et al. (2020), patients underwent various bariatric treatments, and the glucose breath test showed that SIBO was present in a large number of patients (83%) [39]. Interestingly, in a study by Newberry et al. (2016), a hydrogen breath test after lactulose administration was performed to monitor SIBO in patients that had not undergone bariatric treatments. They found that 54.4% of 791 patients had SIBO [44]. That result was curious, particularly in view of the fact that bacterial growth in the intestine is expected to increase after bariatric surgery. That rate was much higher than the rate reported by Mattsson et al. (2017), where the glucose breath test showed that SIBO was detected in only 37% of patients that had not undergone bariatric treatments [45].

In addition to monitoring SIBO with the breath test in patients that had undergone bariatric treatments, Coelho et al. (2019) summarized the results of previous studies that used the hydrogen breath test with lactulose or glucose to determine the prevalence of SIBO in healthy and obese patients. In eight studies, SIBO was detected in 7–40% of healthy patients. In six studies, SIBO was detected in 15–56% of obese patients [34].

Large differences were reported in the prevalence of SIBO among studies that monitored patients for SIBO after RYGB with the lactulose or glucose breath test. Ishida et al. (2007), Sabate et al. (2016), and Coelho et al. (2019) reported SIBO prevalences of 40.5, 40%, and 38.8%, respectively [34,35,38]. In contrast, Andalib et al. (2015) and Lakhani et al. (2008) reported SIBO prevalences of 73% and 100%, respectively [35,36,37,38]. However, Sabate et al. (2017) found that patients that underwent a different bariatric intervention (AGB) showed a significantly lower (10%) SIBO prevalence [38]. These results suggest that differences in SIBO prevalence among patients that underwent the same bariatric procedure were probably not due to a different substrate used in the breath test, although using a different substrate in the studies of Lakhani et al. (2008) and Andalib et al. (2015) resulted in a higher incidence of SIBO compared to the three cases mentioned above (Ishida et al. (2007), Sabate et al. (2016), and Coelho et al. (2019)) which was probably due to the inclusion of patients with unspecified digestive complaints. Moreover, the different results for SIBO in the studies by Mouillot et al. (2020) and Venturi et al. (1994) (83% and 30%, respectively) could have been due to the inclusion of a mixed group of patients that underwent different types of bariatric procedures in one study (RYGB, OAGB, and SG), or using another bariatric approach in comparison with the approach used in other studies (JIB vs. RYGB) [39,43].

In general, differences in SIBO can be caused by the absence of information regarding the prevalence of SIBO in patients before surgery, which could change the interpretation of postoperative data. Last but not least, the results may be influenced by an absence of information regarding the patient’s diet, which could influence the result of the breath test. Moreover, the incidence of SIBO could be related to the use of medications, for example, proton pump inhibitors [34].

The importance of comparing prevalences before and after bariatric treatments was highlighted in two studies on lactose breath tests. Westerink et al. (2020) investigated lactose malabsorption and intolerance in 84 patients before undergoing RYGB and 84 patients after undergoing RYGB with a lactose breath test. Before surgery, 17.9% of patients showed positive results for lactose malabsorption. After surgery, 29.8% of patients showed positive results for lactose malabsorption [41]. Sendiny et al. (2020) studied lactose malabsorption in patients that had not undergone bariatric treatments. They found that 26.7% of 430 patients had positive results for lactose malabsorption [46].

Breath tests can also be used to monitor duodenal pancreatic activity. The ^13^C-mixed triglyceride breath test can detect severe EPI [47,48]. Under stringent conditions, this test can also detect mild to moderate EPI with high sensitivity and specificity [49]. The ^13^C-mixed triglyceride breath test is based on substrate cleavage by pancreatic lipase [50].

Uribarri-Gonzales et al. (2021) used the ^13^C-mixed triglyceride breath test to monitor pancreatic function and the dynamics of digestion and nutritional absorption. They found that EPI occurred in 75% of patients after BPD/DS, in 8.3% of patients after RYGB, and in 4.3% of patients after SG [42]. For comparison, Keller et al. (2014) studied pancreatic function with the ^13^C-mixed triglyceride breath test in patients that did not undergo bariatric treatments. Of the 181 patients studied, 50 (27.6%) had impaired lipolysis and EPI [51].

In addition to conventional breath tests, gas chromatography can be performed to measure the amount of exhaled hydrogen and methane. Mathur et al. (2016) used this approach in patients that underwent RYGB and SG treatments. They found that only 8.3% of patients exhaled both gases, and these patients lost less weight after bariatric surgery than all the other patients. In this publication, the authors mentioned that the presence of both gases, methane and hydrogen, in the case of breath tests, is associated with increased BMI and percentage of body fat in patients [40]. For comparison, the lactulose breath test was used to monitor hydrogen and methane in exhaled air in 11,674 patients that did not undergo bariatric treatments. They showed that relatively few patients (5.5%) had both methane and hydrogen in exhaled air [52].

## 5. Conclusions

Despite the use of breath testing in the context of bariatric treatment, very few studies have been found in the last three decades that specifically address the use of breath testing in patients undergoing various bariatric procedures.

In most of the publications found, the hydrogen breath test using various substrates was used and it was also found that studies dealt with SIBO monitoring in bariatric patients. With one exception, where JIB was used as a bariatric procedure, the studies used RYGB, either alone or in combination with other bariatric approaches.

It can be said that apart from the similarities in the results in the Coelho et al. (2019) [34] and Ishida et al. (2007) [35] studies, where the same breath test and bariatric approaches were used, and SIBO was followed in both cases, the results differed within the framework of studies using the same breath test, the same bariatric approach, and the same process/phenomenon studied (Ishida et al. (2008) [35] vs. Andalib et al. (2015) [37]). This was also the case across studies, which was due to the use of different breath tests, different processes/phenomenon monitored, and also different bariatric interventions, or rather the use of one versus a combination of bariatric interventions.

As mentioned in the discussion, the results of breath tests while monitoring the same process/phenomenon are also quite different when comparing studies on bariatric and non-bariatric patients.

The future use of breath testing in the case of bariatric patients (either before or after surgery) to monitor their condition certainly looks promising, but more studies would are required to obtain more comprehensive results and ideas on how to use breath testing in bariatrics.

## Data Availability

Not applicable.

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
