# Peer review of "Breath Tests Used in the Context of Bariatric Surgery"

_diagnostics, 2022, doi:10.3390/diagnostics12123170_

Round 1
Reviewer 1 Report
The aim of the paper was to present the usefulness of breath tests in the diagnostics of various GI problems in patients subjected to bariatric surgery.
In the introduction an overview of various breath test is announced, but the title of the table 1. limits the content to the bariatrics?. Additionally the Authors should refer to the current quideline on the use of breath test “European guideline on indications, performance, and clinical impact of hydrogen and methane breath tests in adult and pediatric patients…” (2022) .
I would also suggest the change of the order of paragraphs - the history, description of the procedures etc as first, followed by the “Search Strategy and Selection Criteria” and the use of Breath tests in bariatrics.
Discussion
In the discussion the aspects of general use of breath tests and their use in bariatrics are mixed up. Some One of the processes that can be monitored with the breath tests is SIBO. In this breath test, 13C is detected after the administration of 13C-xylose” . The more critical overview of the cited studies in bariatric patients should be presented – methodology, limitations.
Author Response
In the introduction an overview of various breath test is announced, but the title of the table 1. limits the content to the bariatrics?. Additionally the Authors should refer to the current quideline on the use of breath test “European guideline on indications, performance, and clinical impact of hydrogen and methane breath tests in adult and pediatric patients…” (2022).
- Change of the name of the table from "Summary of how breath tests are used in bariatrics" to more general "Current usage of breath tests."
- Text inserted after the first paragraph throughout the article: The usage of breath tests must meet European guideline on indications, performance, and clinical impact of hydrogen and methane breath tests in adult and pediatric patients: European Association for Gastroenterology, Endoscopy and Nutrition, European Society of Neurogastroenterology and Motility, and European Society for Paediatric Gastroenterology Hepatology and Nutrition consensus from 2022.
I would also suggest the change of the order of paragraphs - the history, description of the procedures etc as first, followed by the “Search Strategy and Selection Criteria” and the use of Breath tests in bariatrics.
- Order of paragraphs changed.
Discussion
In the discussion the aspects of general use of breath tests and their use in bariatrics are mixed up. Some One of the processes that can be monitored with the breath tests is SIBO. In this breath test, 13C is detected after the administration of 13C-xylose” . The more critical overview of the cited studies in bariatric patients should be presented – methodology, limitations.
1. The discussion focuses on the comparison of breath test results from studies on bariatric and non-bariatric patients. The results of bariatric patients (various bariatric procedures) are shown in Table 2, where all publications focusing on breath test measurements in bariatric patients over the last 30 years are included. Most publications have focused on SIBO monitoring, which is therefore the focus of most of the discussion. Based on these facts, we consider the discussion to be adequately described.
Reviewer 2 Report
Thank you very much for the opportunity to review the article under the title: Breath Tests used in the Context of Bariatric Surgery.
Non-invasive tests are extremely important in the detection, treatment and monitoring of chronic diseases. They are very well tolerated by patients. Unfortunately, the repeatability and validation of respiratory tests may cause difficulties in the organization of these tests.
it is worth mentioning whether and how body weight affects the results of exhalation tests. In addition, the authors only mentioned the change in the diet of patients after bariatric surgery.
In view of the dramatic change in food intake in the first months after surgery, it is worth expanding this aspect.
the construction of the article in terms of content in the results and conclusions section requires technical improvement
Author Response
it is worth mentioning whether and how body weight affects the results of exhalation tests. In addition, the authors only mentioned the change in the diet of patients after bariatric surgery.
In view of the dramatic change in food intake in the first months after surgery, it is worth expanding this aspect.
1. The presence of both methane and hydrogen on breath testing is associated with increased BMI and percent body fat in humans.
the construction of the article in terms of content in the results and conclusions section requires technical improvement.
1. We think that given the very few publications dealing with the use of breath testing in bariatrics, the results and conclusions are sufficient.